# Cytokine Storms in the Course of COVID-19 and Haemophagocytic Lymphohistiocytosis in Pregnant and Postpartum Women

**DOI:** 10.3390/biom11081202

**Published:** 2021-08-13

**Authors:** Aleksandra Obuchowska, Arkadiusz Standyło, Karolina Obuchowska, Żaneta Kimber-Trojnar, Bożena Leszczyńska-Gorzelak

**Affiliations:** Department of Obstetrics and Perinatology, Medical University of Lublin, 20-090 Lublin, Poland; aobuchowska12@gmail.com (A.O.); karolinaobuchowska99@gmail.com (K.O.); zkimber@poczta.onet.pl (Ż.K.-T.); b.leszczynska@umlub.pl (B.L.-G.)

**Keywords:** cytokine storm, COVID-19, haemophagocytic lymphohistiocytosis, pregnancy, postpartum period

## Abstract

The term ‘cytokine storm’ (CS) applies to a pathological autoimmune reaction when the interactions that lead to cytokine production are destabilised and may even lead to death. CS may be induced by severe acute respiratory syndrome coronavirus 2 (SARS-CoV-2) infection. In this study, we present our analysis of certain pathological processes that induce a CS in pregnant and postpartum women. We draw our attention to the similarities between the severe course of Coronavirus Disease 2019 (COVID-19) and haemophagocytic lymphohistiocytosis (HLH). It is noteworthy that many of the criteria used to diagnose HLH are described as COVID-19 mortality predictors. Cytokine storms are considered to be an important cause of death in patients with the severe course of SARS-CoV-2 infection. Due to the fact that pregnant women are in an immunosuppressive state, viral pulmonary infections are more perilous for them—possible risks include miscarriage, intrauterine growth restriction or birth before the term; sometimes ventilation support is needed. HLH should be considered in pregnant and puerperal women suffering from moderately severe to severe COVID-19 and presenting with: fever unresponsive to antibiotic therapy, cytopenia, hepatitis and hyperferritinaemia. The HLH disorder is rare and difficult to diagnose; however, its early detection could reduce patient mortality.

## 1. Introduction

Pregnancy is a special time in a woman’s life. During this period, physiological changes take place not only in the endocrine system but in the immune system as well. Trophoblasts and the maternal immune system reach a consensus, and the immune response during pregnancy is a combination of the maternal systemic response and the local foetal-placental response. Scientific studies have shown that due to immune changes, also associated with altered cellular immunity, pregnant women are susceptible to severe infectious respiratory diseases [1,2]. Their response to microorganisms varies according to the type of pathogen and stage of pregnancy. Many pathologies developing during pregnancy, e.g., preeclampsia (PE), result from abnormal immune changes. An abnormal response of the maternal immune system to the placenta may be the first pathogenetic stage of PE, followed by a systemic inflammatory response [3].

A cytokine storm (CS) is a form of uncontrolled systemic inflammatory response that may even lead to death. It can be caused by various factors, most often by an infection, systemic diseases (such as rheumatoid arthritis, systemic lupus erythematosus) and, though quite rarely, by anaphylaxis [4]. A CS may also be triggered by medical interventions, such as transplantation or drug administration [5], or induced by SARS-CoV-2 infection [6].

In December 2019, a novel coronavirus 2 (SARS-CoV-2), which triggers a severe respiratory disease in humans, named COVID-19, was first identified in Wuhan, China. Since then, COVID-19 has spread rapidly throughout the world by human-to-human transmission. The elderly, as well as patients with coexisting medical conditions, appear to be most susceptible to COVID-19, and they are also more likely to develop the severe form of the disease as SARS-CoV-2 can lead to acute respiratory distress syndrome (ARDS). The World Health Organization announced the COVID-19 outbreak as a global pandemic in March 2020. This has had a huge effect on the world economy and caused a massive health crisis [7].

In this study, we present our analysis of certain pathological processes that induce a CS in pregnant and postpartum women. We draw our attention to the similarities between the severe course of COVID-19 and HLH.

## 2. Physiological Changes in the Immune System during Pregnancy

Pregnancy is a state in which there are temporary changes in the maternal immune response. These adaptations are made to protect the mother and the foetus from pathogens while avoiding a deleterious immune response against the foetus. Macrophages, dendritic cells, neutrophils and natural killer (NK) cells participate in a coordinated, controlled immune response in early pregnancy, which is needed for implantation and pregnancy progression. The maternal immune T-cell profile shifts from the type 1 T helper cells (Th1) dominance to the type 2 T helper cells (Th2) dominance, which is a physiological state (Figure 1).

Elevated levels of progesterone and oestrogen during pregnancy are responsible for inflecting the immune response by promoting the Th2 and regulatory T cell (T-reg) responses with concurrent suppression of the Th1 and T helper 17 cells (Th17) responses [8]. There are protective benefits of the observed increased levels of Th2 in pregnancy. The changes in the immune system are impelled by the number of cytokines and endocrine factors. Cytokines such as interferon gamma (IFN-γ) and tumour necrosis factor (TNF-α) are expressed by the Th1 subgroup of CD4 T cells, and they are essential in the cytotoxic immune response. On the other hand, cytokines such as interleukin 4 (IL-4), interleukin 5 (IL-5) and interleukin 13 (IL-13), which are involved in the humoral response, are expressed by the Th2 subgroup. The T-reg cells, which are a subset of CD4 T cells, are important regulators of the maternal immune response and foetal tolerance. In order to suppress local immunity, the T-reg cells produce interleukin 10 (IL-10) and transforming growth factor beta (TGF-β). Decreased levels of circulating T-reg cells are observed in women with recurrent miscarriage or PE in comparison to healthy pregnant women [8].

In the third trimester, there is an inflow of immune cells into the myometrium and the production of cytokines intensifies. This results in contraction of the uterus and delivery of the baby.

The mechanisms presented above can prevent the surplus systemic inflammatory reaction and development of life-threatening complications in COVID-19 patients [9].

## 3. Immunological Aspects in COVID-19 and HLH

SARS-CoV-2 is a beta coronavirus most closely related to SARS-CoV. It is reported that the homology between the genome of the SARS-CoV-2 and SARS-CoV is 82% [10]. In order to gain entry to the cells, both viruses use the angiotensin-converting enzyme-related carboxypeptidase (ACE2) receptor. The attachment of the virus with ACE2, as its cellular receptor, triggers internalisation of the complex into the target cell, leading to the down-regulation of the ACE2 [11]. SARS-CoV-2 enters the cell through the ACE2 receptor, the activity of which is increased in normal pregnancy [12] (Figure 2).

Evidence suggests that SARS-CoV-2 driven ACE2 down-regulation leads to an array of complex and intertwined molecular interactions resulting in the observed CS in severe COVID-19 patients [13]. This immune response has been associated with higher intensive care unit (ICU) admissions and mortality among COVID-19 patients [14].

The immune system plays a crucial role in protecting the body against disease; thus, a rapid immune response is important during infection. An initial response of the organism to acute inflammation is an increase in blood flow. This allows the plasma and leukocytes to reach the site of inflammation and activate pro-inflammatory cytokines or chemokines, which in turn leads to the recruitment of the inflammatory cells and an increase in the expression of the pro-inflammatory, antiviral and apoptotic genes [15]. However, if any dysregulation occurs, disease develops. A cytokine storm is an example of such dysregulation of the immune system.

The term ‘cytokine storm’ applies to a pathological autoimmune reaction when the interactions that lead to cytokine production are destabilised. The other cells of the immune system are affected by the positive feedback driven by the overproduced cytokines. It results in producing unrestrained inflammation within the tissues and key organs and induces an autoimmune system attack [14]. The damage is usually removed due to regeneration; however, in a cytokine storm, more serious effects of inflammation are involved, including diffuse alveolar damage or wound fibrosis [16]. Cytokine storms are associated with sepsis and septic shock, influenza, acute respiratory distress and a toxic response to medication. Cytokine storms are considered to be an important cause of death in patients with the severe course of SARS-CoV-2 infection [14].

Cytokines are small proteins involved in the cell signalling pathways. They belong to the immune system and are produced by monocytes, macrophages, neutrophils, B-cells, and T-cells. These molecules regulate the immune reaction and inflammatory responses of the body to infections [16].

Patients suffering from COVID-19 have abnormal laboratory results, i.e., the levels of inflammatory cytokines-interleukin 6 (IL-6), interleukin 1β (IL-1β), TNF-α, procalcitonin, C-reactive protein (CRP) and angiotensin II are increased, highlighting the role of the inflammatory responses in the disease [16]. Increased levels of several cytokines, especially interleukin IL-1β, interleukin 2 (IL-2), IL-6, interleukin 7 (IL-7), IL-10, TNFα and granulocyte colony-stimulating factor (G-CSF) were observed in patients with SARS-CoV-2. The number of total T, B and NK cells was significantly reduced in COVID-19 patients. Furthermore, the CD8+ T cells were skewing toward a senescent phenotype. As a consequence, the CD4+ T, CD8+ T and NK cells presented decreased antiviral cytokine productivity. Cytokines, such as interleukin 12 IL12, IL15 and IL21, which are important for NK-cell activity, were not discovered consistently. Reduced cytotoxic potential was discovered, especially in COVID-19 patients who required intensive care [17,18].

In addition, the CD4+ T cells are activated to become pathogenic Th1 cells and generate G-CSF, thereby augmenting the expression of IL-6 in the CD14+, CD16+ monocytes [14]. IL-1 and IL-6 are the major pro-inflammatory cytokines released by the host in response to a viral infection. It seems that the level of IL-6 is the most important here as higher levels of this interleukin were manifested in patients with the severe course of COVID-19. In addition to this, patients with severe COVID-19 who received mechanical ventilation showed increased levels of IL-6 [11].

According to recent studies, IL-6 may be responsible for severe lung inflammation and pulmonary function disability in patients with severe COVID-19. Moreover, it has been reported that patients with increased baseline levels of IL-6 required prolonged periods of intensive care and specialised treatment [11,16].

It was observed that COVID-19 patients expressed no or low levels of anti-inflammatory factors comparing to the amount of inflammatory agents, which resulted in the immune imbalance and promotion of a cytokine storm. This massive release of pro-inflammatory mediators is characteristic in a group of conditions sharing the same pathogenic mechanism, although with a different aetiology. It is suggested that the reason for ARDS is the perturbated hyperinflammatory response caused by a CS. In COVID-19, a CS is also held accountable for another cause of mortality, i.e., secondary haemophagocytic lymphohistiocytosis (sHLH) [19].

HLH is classified as one of the cytokine storm syndromes (CSS). HLH is a rare, life-threatening disorder. It may be provoked either by a primary genetic defect or associated with several diseases such as infection, haematologic malignancy, rheumatologic illness, immunodeficiency syndromes, autoimmune diseases and drugs. Most of the reported cases were associated with a viral infection, including Epstein-Barr virus (EBV), human immunodeficiency viruses, cytomegalovirus, herpes simplex virus, influenza virus, hepatitis-B-virus and parvovirus B19 [20,21,22]. The methods by which viruses cause HLH have not yet been fully explained. Various factors may play a role. Viruses, particularly DNA viruses, have the ability to modulate the immune response. Herpes viruses, in particular, exploit this technique [23]. They can enhance the host’s vulnerability to HLH by deliberately avoiding effector immune responses and causing a cytokine imbalance. Pathogenic microbes cause excessive T-cell and macrophage proliferation and activation, resulting in dysregulated cellular immunity and Th1/Th2 ratios, excessive Th1 cell activation, secretion of large amounts of cytokines that activate cytotoxic T cells (CTLs) and macrophages, proliferation of a large number of CTLs, and enhanced phagocytosis in macrophages [24]. These pathogenic microbes also stimulate macrophages, DCs, NK cells, and CTLs, resulting in a cytokine storm that destroys target cells. CTLs will be continually activated by proliferative signals and release huge quantities of IFN, causing macrophages to secrete different cytokines and chemokines, culminating in a cascade of uncontrolled inflammatory reactions if immune cell activation is not halted [24].

In the case of CSS caused by pathogenic microorganisms, not only must the diagnostic criteria of HLH-2004 be met, but also there must be evidence of an aetiological diagnosis (positive tests for the presence of virus or microorganism isolation) [24]. Despite extensive diagnostics, in 23–26% of the cases, the aetiological factor remains unknown [25]. HLH is a T-cell-driven hyperinflammatory, hyperferritinemic condition characterised by persistent IFN-γ-dependent stimulation of the Toll-like receptors, antigen-presenting cells and T-cell uncontrolled activation, ultimately leading to a CSS [26].

COVID-19 diagnosis is based on symptoms, epidemiological history, chest imaging and virus nucleic acid tests [27]. Identification of predictive factors may be helpful in conducting the evidence-based treatment of COVID-19. Nevertheless, current knowledge about pathophysiological parameters, especially immunological indicators, is not sufficient to be able to fully understand and explain the mechanism in COVID-19. A cytokine storm may be responsible for the severe course of COVID-19 characterised by alveolar damage and respiratory failure. It has been reported that autopsies of several patients with diffuse alveolar damage within the lungs revealed haemophagocytosis in the pulmonary lymph nodes [28]. The COVID-19 infection may lead to the development of multiple organ disfunction with signs of endothelial derangement [29]. It has been suggested that vascular damage might be a pertinent mechanism, which can sustain the activation of the complement system, and as a result, the inflammatory stimulus might be intensified. As a consequence of vascular damage, the circulating endothelial cells (CECs) level increases rapidly, so endothelial derangement may be detected by reliable markers such as CECs [29]. CECs are erythroid progenitors and precursors with immunosuppressive and/or immunomodulatory properties. These include cell interactions, transformation of growth factor β, reactive oxygen species (ROS) and arginase II and suppression of T cell and B cell functions as well. CECs from COVID-19 patients expressed arginase I/II and ROS, which resulted in global immunosuppression of T cells and impairment of cytokine production, proliferation and degranulation capacities of antigen-specific T cells. CECs from COVID-19 patients also presented a different membrane structure. It can be assumed that the SARS-CoV-2 infection can have consequences, such as the modification of CECs structural components [30].

The immune mechanism of the COVID-19 cytokine storm may be related to virus-induced immunosuppression and NK cell dysfunction [31]. NK cells responses are of enormous significance for antiviral immune responses. A recently published study of COVID-19 patients revealed that severe disease was associated with NKG2C+ NK cell prevalence. After the interaction between CD94/NKG2C and its cellular ligand HLA-E, cytotoxic NK cell responses are activated. As the result of that interaction, pro-inflammatory effector molecules of NKG2C+ cells are released against virus-infected cells [31].

Increased levels of interferon gamma-induced protein 10, monocyte chemotactic protein 3 and IL-1 receptor antagonist associated with the disease severity and fatality have been observed in severe ARDS pneumonia patients [32].

Liu et al. observed that in the patients suffering from SARS-CoV-2, the baseline levels of IL-2, IL-4, TNF-α, and IFN-γ were within the normal range, while IL-10 was slightly increased, whereas IL-6 was significantly increased [14]. Ying Sun et al. revealed that both the serum ferritin and IL-6 levels were significantly elevated in the severe and critically ill groups of patients [33]. Increased IL-6 levels are indicative of the presence of a hyperimmune inflammatory state, and they foretell higher morbidity and mortality. Furthermore, serum ferritin levels can be affected by iron status and may indicate a hyperimmune state [33]. In addition to this, the total T lymphocytes, CD4+, CD8+ T lymphocytes and B lymphocyte counts were found to be significantly decreased in severe and critically ill patients on their admission. These results show that SARS-CoV-2 has a negative effect on T-cell mediated immunity [34].

## 4. The Course of COVID-19 in Pregnant and Postpartum Women

Due to the fact that pregnant women are in an immunosuppressive state, viral pulmonary infections are more perilous for them than infections of bacterial origin, and they are particularly susceptible to respiratory pathogens and severe pneumonia. Physiologic adaptive changes during pregnancy, including diaphragm elevation, increased oxygen consumption and changes in the mucous membrane of the respiratory tract, render them intolerant to hypoxia. Pneumonia is a type of lung disease that can be very dangerous for pregnant women—possible risks include miscarriage, intrauterine growth restriction or birth before the term; sometimes ventilation support is needed [35].

Various findings on the relationship between COVID-19 and infection severity in pregnant women have been reported. A review of the specialist literature indicates that no significant differences in the clinical symptoms in the group of pregnant women infected with SARS-CoV-2 in comparison with the overall patient population have been found so far.

The severity of the disease in the general population and in pregnant women appears to be comparable (mild—86%; severe—9.3%, critical—4.7%) [36].

According to the literature reports, in the third trimester of pregnancy, the main features of symptomatic COVID-19 were fever (40–68%) and coughing (34–41%), dyspnoea (12%) and lymphopenia (33%) [37,38]. Both leukocytosis and leukopenia are common in pregnant COVID-19 patients. The less frequent symptoms include headaches, excessive sputum production and changes in the sense of smell and taste. The symptoms do not appear to be dependent on the gestational age [37]. Generally, most of the COVID-19 patients have a fairly mild or asymptomatic course of the disease; only about 15% of cases are severe (oxygen support is needed), and 5% of cases, where mechanical ventilation is required, are critical. In the study of Turan et al., 76.5% of patients presented a mild course of disease, in 15% the course was severe, and 7.7% presented a critical stage of the disease at the time of admission [39]. It has been reported that women with a critical state of COVID-19 at the time of admission account for the vast majority of deaths secondary to COVID-19 [39]. Among pregnant women with the COVID-19 symptoms, 4.99% were admitted to the ICU [40]. Maternal mortality was 1–1.3%, and the rate of severe pneumonia ranged from 0 to 14% [40,41]. It has been found that COVID-19 increased the risk of several frequent pregnancy conditions, including PE and pulmonary embolism [37]. A high body mass index, PE, and pre-existing diabetes have been associated with the severe course of COVID-19. Moreover, existing comorbidities in pregnant women were a major risk factor for ICU admission and mechanical ventilation. Furthermore, a severe coexisting disease was found to be the main cause of death in pregnant women infected with COVID-19 [37,39,41].

It has been noted that decreased levels of erythrocytes and lymphocytes as well as increased levels of alanine aminotransferase and CRP were the main predictors of the severe course of COVID-19 in pregnant women [42].

In their systematic review, Alloney et al. made an observation that the overall risk of preterm birth was 17% and, according to them, pregnant women with COVID-19 were at a higher risk of any preterm birth compared to women without COVID-19 [37]. Premature rupture of membranes occurred in 2.5% of pregnancies [40]. Preterm births occurred in women with the severe course of disease, mostly as a result of early delivery for maternal indications, although there are sporadic reports of spontaneous preterm births [43].

In the analysed literature, infection of a pregnant woman with SARS-CoV-2 is associated with an increased percentage of pregnancy completion by caesarean section in relation to the general population [38]. Caesarean section accounted for 92% of all COVID-19-pregnancy deliveries [10]. In the study of Ferazzi et al., 57.3% of pregnant women underwent elective caesarean section and 34.9% had vaginal delivery [44].

In most cases, the caesarean section was decided during delivery. The choice of this method was argued by foetal distress. Nevertheless, natural births were not associated with worse outcomes [38].

In postpartum women, there is no strong correlation between COVID-19 severity and disadvantageous perinatal outcomes. If there were admissions at the neonatal ICU, 33% of the neonates were born to women with COVID-19 and most often presented with such symptoms as shortness of breath, fever, thrombocytopenia, abnormal liver function, tachycardia, vomiting, and pneumothorax. It appears that the rates of stillbirth and neonatal death are low in women with suspected or confirmed COVID-19 [37,45].

The rate of maternal complications seems to be comparable to non-pregnant adults. In the study of Matar et al., pregnant women infected with COVID-19 have similar clinical characteristics and outcomes as the non-pregnant population, and there is little evidence confirming vertical transmission [46]. According to one theory, immunologic adaptations in pregnancy preventing the mothers from rejecting the foetus, a foreign entity containing paternal antigens, may also be helpful in mounting a less robust immune response to the virus, consequently leading to less destructive effects on the body. Another theory is that pregnancy-related organ adaptive changes may be protective against the virus [46].

Current specialist literature provides information that the majority of clinical manifestations are detected in the late third trimester. Thus, it may not be the mere virus causing severe condition but comorbidities as well. In light of this, it appears that physicians should be alert to the possibility of a severe course of COVID-19 in pregnant women who present risk factors linked to maternal morbidity. Therefore, a proper follow-up of such patients is necessary.

## 5. The Course of HLH in Pregnant and Postpartum Women

Pregnancy is a physiological process, and the HLH syndrome is rarely diagnosed in pregnant women. Cytotoxic T cells, macrophages and NK cells play an essential role in HLH. Excessive activation and proliferation of T-cells and macrophages, and secretion of large amounts of cytokines, may be responsible for multiple organ failure, including the liver, brain and bone marrow, which can be potentially fatal [47]. The levels of cytokines such as TNF-α, IL-1β, IL-6, IL-10, and IFN-γ increase in HLH [48].

Typical HLH symptoms include fever (which is unresponsive to antibiotic therapy and sometimes hectic), splenomegaly (69%), hepatomegaly (67%), pulmonary involvement (42%), skin lesions (25%), neurological involvement (25%) and lymphadenopathy [22]. Fever occurs in most diagnosed HLH cases (95,7%) [22,25]. Interestingly, many pregnant women have been diagnosed with fatty liver disease and jaundice. However, following HLH remission, liver function usually returns to normal [21]. Because of ambiguous symptoms and rare occurrence, the diagnosis of HLH is rather complicated. While making a diagnosis, other diseases, such as infection, sepsis, acute fatty liver of pregnancy, acute cholecystitis, Stills disease, HELLP syndrome and aplastic anaemia, should be excluded.

In order to diagnose HLH, either molecular diagnostics consistent with HLH must be performed or five of the eight diagnostic criteria for HLH must be fulfilled, i.e., splenomegaly, fever, cytopenia (affecting two or more of three lineages in the peripheral blood), hypofibrinogenaemia and/or hypertriglyceridaemia, elevated levels of ferritin, haemophagocytosis in the bone marrow/spleen/lymph nodes, low or absent NK cell activity and increased levels of soluble CD25 (interleukin [IL]-2 receptor), or both [26]. 

In the study of Ramos Casals et al., anaemia and thrombocytopenia occur in 80–90% of adult HLH patients, and leukopenia is present in almost 70% of them [22]. Almost 60% of patients have coagulation disorders; hypofibrinogenaemia and raised D-dimer levels are reported in 50% of cases. In the mentioned study, 80% of patients had elevated levels of liver enzymes [22].

Fardet et al. developed a scoring system (HScore) that is useful in recognising HLH [49]. However, these criteria apply mainly to primary HLH and to a lesser extent to sHLH [50]. Hyperferritinaemia is the most characteristic of HLH, and low serum levels of ferritin preclude its recognition. Macrophage activation syndrome is a type of HLH that most often coexists with rheumatic diseases, such as Still’s disease, systemic lupus erythematosus, dermatomyositis, juvenile idiopathic arthritis (JIA) or systemic sclerosis. Despite the fact that the complete pathways are unknown, decreased natural killer (NK) cell activity, elevated soluble interleukin-2 receptor (sIL-2R), and broadly excessive cytokine production appear to be elements of MAS pathogenesis [51]. Rheumatic diseases can have different courses during pregnancy. It is well known that autoimmune diseases can worsen during pregnancy (near the end of the third trimester). Active disease or flare-ups during pregnancy can have a severe impact on foetal health and the outcome of the pregnancy [52]. Among rheumatic disorders, HLH occurs most frequently in systemic juvenile idiopathic arthritis. In pregnant women with active JIA disease, 57–94% experienced relief of symptoms, and around 50% reported an increased disease activity between 3 and 12 months postpartum. According to studies, 35–70% of women with SLE may develop a flare during pregnancy (especially in the third trimester of pregnancy) or from 3 to 12 months postpartum. High disease activity before conception increases the likelihood of an exacerbation of the disease in pregnancy [53].

HLH in the group of pregnant and postpartum women accounts for high morbidity and mortality among them. Therefore, effective treatment is needed in order to reduce mortality and prevent ICU admission in this population. For this reason, it is important to diagnose the disease timely and implement appropriate treatment promptly. No treatment scheme has been developed for pregnant women so far. However, the HLH-2004 protocol is usually used at the beginning of treatment. The protocol includes therapy with dexamethasone 10 mg/m^2^ daily, etoposide 150 mg/m^2^ twice a week and cyclosporine, followed by dexamethasone intravenous pulses [54]. Steroids are used as the first line of treatment. The use of etoposide, which is a potential teratogenic drug, is controversial in the case of pregnant women. Etoposide might also induce secondary malignancies, especially acute myeloid leukaemia (the risk is estimated at 0.3 to 0.4%) [55]. For these reasons, it is better to use steroids as the first line therapy and intravenous immunoglobulins or cyclosporine in steroids-resistant patients [21,26]. If the central nervous system is affected, intrathecal methotrexate infusions can be used [54].

## 6. Associations between HLH and COVID-19

Recently, some similarity has been noticed between moderate and severe COVID-19 and HLH (Table 1).

Treatment of patients with the severe course of COVID-19 is difficult because the only treatment option available for ARDS is symptomatic treatment. The HLH-2004 protocol has been successfully used in HLH triggered by EBV, HBV and influenza A virus subtype (A/H1N1) infections [55]. These observations suggest that this therapy could be effective in the treatment of COVID-19 patients with the severe course of the disease.

It is noteworthy that many of the criteria used to diagnose HLH are described as COVID-19 mortality predictors [56]. A relationship has been observed between the median time from the onset of COVID-19 symptoms to ARDS (which was 8.0–14.0 days) and HLH development during A/H1N1 infection. Hamizi et al. suggest that moderately severe to severe COVID-19 might be a form of sHLH [56]. A cytokine storm has been reported in the ICU patients suffering from SARS-CoV-2 infection [57]. Dimopoulos et al. identified secondary HLH associated with COVID-19 thanks to the use of HScore. On this basis, Anakinra, which turned out to be effective, was included in the treatment [20]. Patients were treated with 200 mg Anakinra intravenously for eight hours every day for a week or 300 mg intravenously once daily for 4 days (high-dose intravenous anakinra is used off-label for the treatment of hyperinflammatory conditions). Anakinra neutralises the biological activity of interleukin-1α and interleukin-1β, which is an important part of the hyperinflammatory response, and it works by competitive inhibition of interleukin type I receptor binding (IL-1RI). Anakinra is supposed to prevent the progression of respiratory failure, and it reduces the need for mechanical ventilation. This medication has been observed to cause a rapid clinical improvement within 24 h and following its administration, oxygen demand decreased, serum ferritin levels, aspartate aminotransferase (AST) and CRP decreased as well [20]. Cavalli et al. found that the use of the IL-1 blockade with high-dose Anakinra brought about mortality reduction by 10% within 21 days [58].

However, some research suggests that the use of HScore to identify the COVID-19 cytokine storm is of limited value [59,60].

Treatment with etoposide and dexamethasone is aimed at suppressing the cytokine storm, and therefore, it may be potentially effective in the treatment of patients with severe COVID-19. It has been shown that etoposide removes activated T lymphocytes and effectively suppresses a cytokine storm. However, no direct anti-inflammatory effect on macrophages or dendritic cells and no deletion of quiescent naive or memory T cells have been reported. Despite the risks associated with administration of etoposide, it could reduce mortality and the nervous system complications in patients with the severe course of COVID-19 [56]. It is noteworthy that in the case of sHLH and sepsis, mortality is very high (66.7%) [61]. The sHLH occurs in 3.7–4.3% of sepsis cases [15]. Thus, its use in pregnant women in interdisciplinary teams should be carefully considered.

Clinical and laboratory factors indicative of the disease severity are: advanced age, sputum production, dyspnoea, elevated neutrophil count, AST, lactate dehydrogenase and gamma-glutamyl transferase levels, elevated CRP and high serum ferritin level, increased IL-6 and D-dimer [33,62].

Severe COVID-19 cases may benefit from the IL-6 pathway inhibition, given the associated cytokine release syndrome (CRS) and sHLH-like serum cytokine elevations. The level of inflammatory cytokines and chemokines such as IL-1a/β, IP-10, MCP-1 is increased as well. Severe cases show elevation of TNFα, IL-1, IL-6, IL-18, IL-8, IL-10, MCP-1 and MIP-1A, which can lead to serious pulmonary tissue damage. Additionally, IL-1 has also been linked to the expression of thromboxane-A2 in COVID-19 patients, and as a result, platelet activation and aggregation are increased, which enhances the risk of thrombus development [63]. Therapeutics based on suppressing CRS, such as tocilizumab, enter clinical trials before they are ready to fight COVID-19 [20]. The immediate goal of IL-6 antagonism is to ameliorate severe COVID-19 cases so that requirements for advanced care are minimised [13].

Similarly, the majority of patients with COVID-19 present with high fever (observed in 44% of patients upon presentation, and subsequently, in 64.5–99% patients), cough (45–82%), dyspnoea (6.5–63.5%), and myalgia or fatigue (11–70%). Some patients also show liver damage (transaminase activity), lymphocytopenia, and rapidly progressing multi-organ failure [50].

## 7. Conclusions

It is worth noticing that pregnant women usually suffer from a mild form of COVID-19, and many of them may be asymptomatic carriers of the SARS-CoV-2. However, some women have been reported to have had a severe course of the disease that required ICU treatment. The prognosis in these patients was poor, which makes it important to incorporate a more advanced specialist treatment. Cytokine storms have been observed in moderate to severe COVID-19 cases. Severe cases were associated with significantly increased levels of ferritin and IL-6, which suggested that in some pregnant women with severe COVID-19, sHLH may be diagnosed. The use of HScore and careful analysis of laboratory test results indicative of a possible cytokine storm in a patient with severe COVID-19, i.e., increased serum ferritin, worsening of bicytopenia or pancytopenia, and increased organ damage rates, appear to be of great importance. This will enable timely detection of the secondary HLH and prompt implementation of appropriate treatment effective enough to reduce maternal mortality as well as mortality of their children. HLH should be considered in pregnant and puerperal women suffering from moderately severe to severe COVID-19 and presenting with: fever unresponsive to antibiotic therapy, cytopenia, hepatitis and hyperferritinaemia. Determining the level of ferritin can significantly fast-track making an accurate diagnosis, especially in intensive care units.

The complicated phenomenon of maternal immunological tolerance to foetal antigens enables the development of an antigenically foreign foetus in the uterus despite the existence of the maternal immune system capable of rejection. The maternal immune T-cell profile shifts from the Th1 dominance to the Th2 dominance. Pregnant women are more susceptible to infections, and viral infections may be very dangerous for them. It is important to clarify whether the cytokine storm present in the context of pregnancy promotes the development of HLH.

Pregnant women are more susceptible to infections, and viral infections may be very dangerous for them. The severity of the viral infection may be dependent on existing comorbidities as well (asthma, gestation diabetes, obesity). The severity of COVID-19 disease is an indication for cesarean section because of possible respiratory complications. Even though the mechanism of COVID-19 is not fully understood, a cytokine storm is held accountable for a more severe course of the disease. COVID-19 patients are reported to have frequently shown abnormal laboratory test results indicative of a cytokine storm, including elevated levels of serum ferritin and IL-6, which makes it possible to diagnose HLH as well. The major feature of HLH is hyperferritinaemia, which is also correlated with poor prognosis in COVID-19 patients. SARS-CoV-2 infection can be a trigger of the disorder, but not all patients suffering from COVID-19 develop HLH. The relationship between HLH and COVID-19 involves mortality predictors for COVID-19, used as criteria to diagnose HLH. The HLH disorder is rare and difficult to diagnose; however, its early detection could reduce patient mortality.

It is important to clarify whether the cytokine storm present in the context of pregnancy promotes the development of HLH. Moreover, future research should focus on elucidating the complex relationship between HLH in pregnancy and COVID-19 infection. The current state of knowledge and the dilemmas regarding this phenomenon have been described by us above.

## Figures and Tables

**Figure 1 biomolecules-11-01202-f001:**
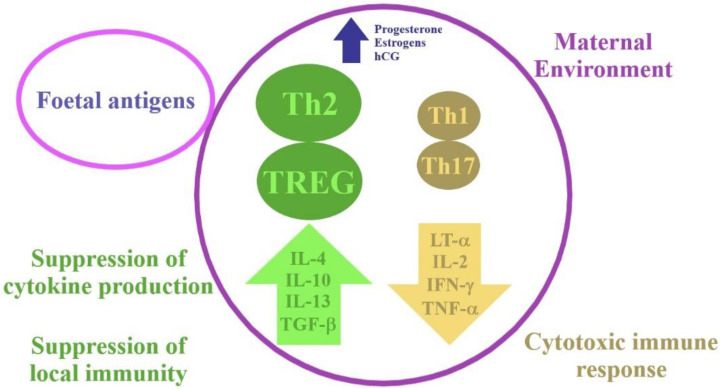
Physiological changes in the immune system in early pregnancy. hCG-human chorionic gonadotropin; IFN-γ-interferon gamma; IL-2-interleukin 2; IL-4-interleukin 4; IL-10-interleukin 10; IL-13-interleukin 13; LT-α-lymphotoxin-alpha; TGF-β-transforming growth factor beta; Th1-type 1 T helper cells; Th2-type 2 T helper cells; Th17-T helper 17 cells; TNF-α-tumour necrosis factor; TREG-regulatory T cell.

**Figure 2 biomolecules-11-01202-f002:**
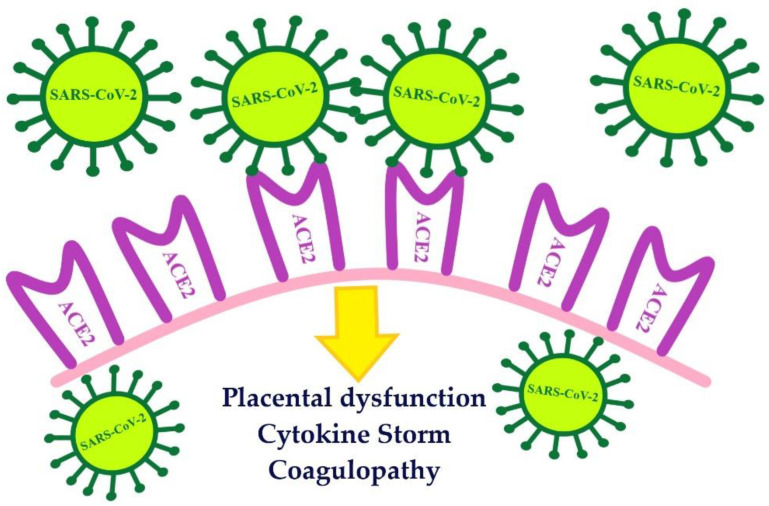
The number of ACE2 receptors increases during early and mid-pregnancy. ACE2 are receptors for the entry of SARS-CoV-2 into placental cells. ACE2-angiotensin-converting enzyme-related carboxypeptidase; SARS-CoV-2-severe acute respiratory syndrome coronavirus 2.

**Table 1 biomolecules-11-01202-t001:** Moderately severe to severe COVID-19 as a form of sHLH. sHLH—secondary haemophagocytic lymphohistiocytosis; COVID-19—coronavirus disease 2019; HLH—haemophagocytic lymphohistiocytosis; IL-6—interleukin 6.

Moderately Severe to Severe COVID-19	sHLH
**prevalence in pregnant women—rarely diagnosed**	
**aetiology**	
viral infection	viral infection in most cases
**hyperferritinaemia**	
marker of poor prognosis in COVID-19 patients	the most characteristic laboratory result of HLH
**treatment**	
treatment aimed at suppressing the cytokine storm	
**high mortality**	
**increased levels of IL-6**	

## Data Availability

MDPI Research Data Policies.

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
