# Peer review of "Cytokine Storms in the Course of COVID-19 and Haemophagocytic Lymphohistiocytosis in Pregnant and Postpartum Women"

_biomolecules, 2021, doi:10.3390/biom11081202_

Round 1

Reviewer 1 Report

In this review, the authors did analysis on certain pathological processes that induce cytokine storm in pregnant and postpartum women. The authors described the similarities between the severe course of Coronavirus Disease 2019 (COVID-19) and haemophagocytic lymphohistiocytosis including the similarities in the criteria used to diagnose haemophagocytic lymphohistiocytosis that are also described as COVID-19 mortality predictors. The haemophagocytic lymphohistiocytosis disorder is rare and difficult to diagnose, however its early detection could reduce patient mortality. This review article provides a comprehensive overview on HLH in pregnant and puerperal women suffering from moderately severe to severe COVID-19.

Author Response

Dear Reviewer 1,
We would like to resubmit our manuscript entitled “Cytokine storms in the course of COVID-19 and haemophagocytic lymphohistiocytosis in pregnant and postpartum women”. In this review, the authors did analysis on certain pathological processes that induce cytokine storm in pregnant and postpartum women. The authors described the similarities between the severe course of Coronavirus Disease 2019 (COVID-19) and haemophagocytic lymphohistiocytosis including the similarities in the criteria used to diagnose haemophagocytic lymphohistiocytosis that are also described as COVID-19 mortality predictors. The haemophagocytic lymphohistiocytosis disorder is rare and difficult to diagnose, however its early detection could reduce patient mortality. This review article provides a comprehensive overview on HLH in pregnant and puerperal women suffering from moderately severe to severe COVID-19.
Thank you very much for finding the time to read our manuscript. Thank you for considering our manuscript.
Yours faithfully,
Arkadiusz Standyło
Chair and Department of Obstetrics and Perinatology, Medical University of Lublin, 20-090 Lublin, Poland
Tel: +48-81-7244-769
E-mail: [email protected]

Reviewer 2 Report

In this article, the authors provide a fairly reasonable overview for cytokine release syndrome (cytokine storm), with an emphasis on Covid19 in pregnancy.  I think overall, the authors have done a reasonable job, although there are certain areas that need expansion.  There are also some text/English changes that need to be rewritten.  

Although there are good sections emphasized, I have specific comments to the authors:

  1. In the immunological changes during pregnancy, I think a larger description is required.Specifically, I would like the authors to discuss the role of circulating erythroid pre-cursors (CECs) and NK cells which play an important immunoregulatory role.  This is important as these components are dysregulated in COVID-19.  Particular emphasis should describe the protective roles of HLA-E and HLA-G which are important in immunoregulating NK cells in pregnancy.
  2. In the COVID-19 in pregnancy, the authors should include the dysregulated CECs as a potential explanation (see DOI 1016/j.stemcr.2021.04.001, 10.1002/ajh.25881). They should also discuss the role of HLA-E (doi.org/10.1038/s41436-020-01077-7).

  1. In the NK cell dysfunction with severe COVID-19, two more relevant article citations are (1182/bloodadvances.2020002650, 10.1172/JCI138554) and the article discusses CRS in great detail (10.3390/ijms21176351). These articles really should be cited.

  1. It is well established that autoimmune diseases can flare in pregnancy (near the end of the third trimester). Can the authors comment on similar dysregulated mechanisms in rheumatic diseases, and patients with pregnancy that develop severe outcomes?

Author Response

Dear Reviewer 2,
We would like to resubmit our manuscript entitled “Cytokine storms in the course of COVID-19 and haemophagocytic lymphohistiocytosis in pregnant and postpartum women”. We appreciate your valuable remarks and hope that the quality of our manuscript is going to meet your expectations now that we have made some suggested alternations.
We have rearranged our paper in accordance with your valuable comments and suggestions. The manuscript has been checked by a native-speaker. In this article, the authors provide a fairly reasonable overview for cytokine release syndrome (cytokine storm), with an emphasis on Covid19 in pregnancy. I think overall, the authors have done a reasonable job, although there are certain areas that need expansion. There are also some text/English changes that need to be rewritten. Thank you very much for your valuable and thoughtful comments. We find all your remarks spot on therefore we have made a point-by-point correction of the manuscript according to your suggestions. In the immunological changes during pregnancy, I think a larger description is required. Specifically, I would like the authors to discuss the role of circulating erythroid pre-cursors (CECs) and NK cells which play an important immunoregulatory role. This is important as these components are dysregulated in COVID-19. Particular emphasis should describe the protective roles of HLA-E and HLA-G which are important in immunoregulating NK cells in pregnancy.
Thank you very much for your valuable and highly perceptive remarks. We have expanded the issue of the immunological changes during pregnancy. We have presented the role of circulating erythroid pre-cursors (CECs) and NK cells as follows:
“The COVID-19 infection may lead to development of multiple organ disfunction with signs of endothelial derangement [29]. It has been suggested that vascular damage might be a pertinent mechanism which can sustain the activation of complement system and as a result the inflammatory stimulus might be intensified. As a
consequence of vascular damage the circulating endothelial cells (CECs) level increase rapidly so endothelial derangement may be detected by reliable markers such as CECs [29]. CECs are erythroid progenitors and precursors with a immunosuppressive and/or immunomodulatory properties. These include cell interactions, transformation of growth factor β, reactive oxygen species (ROS) and arginase II and suppression of T cell and B cell functions as well. CECs from Covid-19 patients expressed arginase I/II and ROS, which resulted in global immunosuppression of T cell and impairment of cytokine production, proliferation and degranulation capacities of antigen-specific T cells. CECs from COVID-19 patients presented also different membrane structure. It can be assumed that SARS-CoV-2 infection can have consequences as CECs structural components modification [30].”
References:
29. Nizzoli, M.E.; Merati, G.; Tenore, A.; Picone, C.; Consensi, E.; Perotti, L.; Ferretti, V.V.; Sambo, M.; Sabatino, A.D.; Iotti, G.A.; Arcaini, L.; Bruno, R.; Belliato, M. Circulating Endothelial Cells in COVID-19. American Journal of Hematology 2020, 95, 187–188.
30. Shahbaz, S.; Xu, L.; Osman, M.; Sligl, W.; Shields, J.; Joyce, M.; Tyrrell, D.L.; Oyegbami, O.; Elahi, S. Erythroid Precursors and Progenitors Suppress Adaptive Immunity and Get Invaded by SARS-CoV-2. Stem Cell Reports 2021, 16, 1165–1181.
In the COVID-19 in pregnancy, the authors should include the dysregulated CECs as a potential explanation (see DOI 1016/j.stemcr.2021.04.001, 10.1002/ajh.25881). They should also discuss the role of HLA-E (doi.org/10.1038/s41436-020-01077-7).
Following your advice, we have expanded the topics as follows:
“The immune mechanism of the COVID-19 cytokine storm may be related to virus-induced immunosuppression and NK cell dysfunction [31]. NK cells responses are of enormous significance for antiviral immune responses. In recently published study in
COVID-19 patients was revealed that severe disease was associated with NKG2C+ NK cells prevalence. After interaction between CD94/NKG2C and its cellular ligand HLA-E cytotoxic NK cell responses are activated. As the result of that interaction a pro-inflammatory effector molecules of NKG2C+ cells are released against virus-infected cells [31].”
References:
31. Vietzen, H.; Zoufaly, A.; Traugott, M.; Aberle, J.; Aberle, S.W.; Puchhammer-Stöckl, E. Deletion of the NKG2C Receptor Encoding KLRC2 Gene and HLA-E Variants Are Risk Factors for Severe COVID-19. Genet Med 2021, 23, 963–967.
In the NK cell dysfunction with severe COVID-19, two more relevant article citations are (1182/bloodadvances.2020002650, 10.1172/JCI138554) and the article discusses CRS in great detail (10.3390/ijms21176351). These articles really should be cited.
We agree that these publications are very important and valuable. In the current version we added them in references and we have expanded manuscript as follows:
“The number of total T, B and NK cells were significantly reduced in COVID-19 patients. Furthermore, the CD8+ T cells were skewing toward a senescent phenotype. As a consequence the CD4+ T, CD8+ T and NK cells presented decreased antiviral cytokine productivity. Cytokines like interleukin 12 IL12, IL15 and IL21 which are important for NK-cell activity were not discovered consistently. Reduced cytotoxic potential was discovered especially in COVID-19 patients which required intensive care [17,18].”
[…]
“The level of inflammatory cytokines and chemokines such as IL-1a/β, IP-10, MCP-1 is increased as well. Severe cases show elevation of TNFα, IL-1, IL-6, IL-18, IL-8, IL-10,
MCP-1 and MIP-1A, which can lead to serious pulmonary tissue damage. Additionally IL-1 has also been linked to the expression of thromboxane-A2 in COVID-19 patients, as a result platelet activation and aggregation is increased which enhances the risk of thrombus development [63].”
References:
17. Mazzoni, A.; Salvati, L.; Maggi, L.; Capone, M.; Vanni, A.; Spinicci, M.; Mencarini, J.; Caporale, R.; Peruzzi, B.; Antonelli, A.; Trotta, M.; Zammarchi, L.; Ciani, L.; Gori, L.; Lazzeri, C.; Matucci, A.; Vultaggio, A.; Rossi, O.; Almerigogna, F.; Parronchi, P.; Fontanari, P.; Lavorini, F.; Peris, A.; Rossolini, G.M.; Bartoloni, A.; Romagnani, S.; Liotta, F.; Annunziato, F.; Cosmi, L. Impaired Immune Cell Cytotoxicity in Severe COVID-19 Is IL-6 Dependent. J Clin Invest 2020, 130, 4694–4703.
18. Osman, M.; Faridi, R.M.; Sligl, W.; Shabani-Rad, M.T.; Dharmani-Khan, P.; Parker, A.; Kalra, A.; Tripathi, M.B.; Storek, J.; Cohen Tervaert, J.W.; Khan, F.M. Impaired Natural Killer Cell Counts and Cytolytic Activity in Patients with Severe COVID-19. Blood Advances 2020, 4, 5035–5039.
63. van Eeden, C.; Khan, L.; Osman, M.S.; Cohen Tervaert, J.W. Natural Killer Cell Dysfunction and Its Role in COVID-19. International Journal of Molecular Sciences 2020, 21, 6351. It is well established that autoimmune diseases can flare in pregnancy (near the end of the third trimester). Can the authors comment on similar dysregulated mechanisms in rheumatic diseases, and patients with pregnancy that develop severe outcomes?
Thank you very much for your valuable and highly perceptive remarks. Following your advice, we have presented these aspects as follows:
“Macrophage activation syndrome is a type of HLH that most often coexists with rheumatic diseases, such as Still's disease, systemic lupus erythematosus,
dermatomyositis, juvenile idiopathic arthritis (JIA) or systemic sclerosis. Despite the fact that the complete pathways are unknown,, decreased natural killer (NK) cell activity, elevated soluble interleukin-2 receptor (sIL-2R), and broadly excessive cytokine production appear to be elements of MAS pathogenesis [51]. Rheumatic diseases can have different courses during pregnancy. It's well known that autoimmune diseases can worsen during pregnancy (near the end of the third trimester). Active disease or flare-ups during pregnancy can have a severe impact on foetal health and the outcome of the pregnancy [52]. Among rheumatic disorders, HLH occurs most frequently in systemic juvenile idiopathic arthritis. Pregnant women with active JIA disease 57-94% experienced relief of symptoms and around 50% reported an increased disease activity between 3 and 12 months postpartum. According to studies, 35–70% of women with SLE may develop a flare during pregnancy (especially in the third trimester of pregnancy) or from 3 to 12 months postpartum. High disease activity before conception increases the likelihood of an exacerbation of the disease in pregnancy [53].”
References:
51. Peters, A.T.; Prickett, M.H. Adult-Onset Still’s Disease Presenting as Macrophage-Activation Syndrome With Critical Illness in the Third Trimester of Pregnancy: A Case Report. Crit Care Explor 2021, 3, 440.
52. Atteritano, M.; David, A.; Bagnato, G.; Beninati, C.; Frisina, A.; Iaria, C.; Bagnato, G.; Cascio, A. Haemophagocytic Syndrome in Rheumatic Patients. A Systematic Review. Eur Rev Med Pharmacol Sci 2012, 16, 1414–1424.
53. Ko, H.S.; Ahn, H.Y.; Jang, D.G.; Choi, S.K.; Park, Y.G.; Park, I.Y.; Lee, G.; Park, S.H.; Shin, J.C. Pregnancy Outcomes and Appropriate Timing of Pregnancy in 183 Pregnancies in Korean Patients with SLE. Int J Med Sci 2011, 8, 577–583.
We would like to take this opportunity to thank you for all the valuable and highly perceptive remarks which have definitely made a substantial contribution to the quality of our paper.
Yours faithfully,
Arkadiusz Standyło
Chair and Department of Obstetrics and Perinatology, Medical University of Lublin, 20-090 Lublin, Poland
Tel: +48-81-7244-769
E-mail: [email protected]

Reviewer 3 Report

Obuchowska and collaborators try to explain the relationship between the haemophagocytic lymphohistiocytosis (HLH) in pregnancy with the COVID-19 and cytokines storm in the present manuscript. The topic is relevant because the complication in pregnancy during COVID-19 is a little-explored field, and new information is necessary to understand the pathogenesis of viral infection in pregnant women. However, the complexity of COVID-19 in the context of systemic and the pathogenesis with the storm cytokines requires delving into the context of pregnancy since an exacerbated immune response is associated with an adverse outcome or abortions. For which I recommend the authors the following points.
+ HLH is associated with the inflammatory responses, and maybe the cytokine storm is the activator of this process.  However, the authors should delve deeper into how viruses could exert HLH due to the cytokine storm.
 *The clinical evidence of covid and HLH should be reflected or summarized in a table that helps the reader understand this possible association.
*One of the problems in the manuscript is if the cytokine storm is present in the context of pregnancy, has this phenomenon been described by other viral infections and thus promoving a pathological effect such as HLH ???
*Finally, future ideas or questions in the field must be mentioned to show whether there is a real relationship of HLH in pregnancy during a COVID-19 infection.

Author Response

Dear Reviewer 3,
We would like to resubmit our manuscript entitled “Cytokine storms in the course of COVID-19 and haemophagocytic lymphohistiocytosis in pregnant and postpartum women”. We appreciate your valuable remarks and hope that the quality of our manuscript is going to meet your expectations now that we have made some suggested alternations.
We have rearranged our paper in accordance with your valuable comments and suggestions. The manuscript has been checked by a native-speaker. Obuchowska and collaborators try to explain the relationship between the haemophagocytic lymphohistiocytosis (HLH) in pregnancy with the COVID-19 and cytokines storm in the present manuscript. The topic is relevant because the complication in pregnancy during COVID-19 is a little-explored field, and new information is necessary to understand the pathogenesis of viral infection in pregnant women. However, the complexity of COVID-19 in the context of systemic and the pathogenesis with the storm cytokines requires delving into the context of pregnancy since an exacerbated immune response is associated with an adverse outcome or abortions. For which I recommend the authors the following points.
Thank you very much for your valuable and thoughtful comments. We find all your remarks spot on therefore we have made a point-by-point correction of the manuscript according to your suggestions. HLH is associated with the inflammatory responses, and maybe the cytokine storm is the activator of this process. However, the authors should delve deeper into how viruses could exert HLH due to the cytokine storm.
Thank you very much for your comments. Following your advice, we have presented how viruses could exert HLH due to the cytokine storm. In the current version we have expanded manuscript as follows:
“The methods by which viruses cause HLH have not yet been fully explained. Various factors may play a role. Viruses, particularly DNA viruses, have the ability to modulate the immune response. Herpes viruses, in particular, exploit this technique [23]. They
can enhance the host's vulnerability to HLH by deliberately avoiding effector immune responses and causing a cytokine imbalance. Pathogenic microbes cause excessive T-cell and macrophage proliferation and activation, resulting in dysregulated cellular immunity and Th1/Th2 ratios, excessive Th1 cell activation, secretion of large amounts of cytokines that activate a cytotoxic T cells (CTLs) and macrophages, proliferation of a large number of CTLs, and enhanced phagocytosis in macrophages [24]. These pathogenic microbes also stimulate macrophages, DCs, NK cells, and CTLs, resulting in a cytokine storm that destroys target cells. CTLs will be continually activated by proliferative signals and release huge quantities of IFN, causing macrophages to secrete different cytokines and chemokines, culminating in a cascade of uncontrolled inflammatory reactions if immune cell activation is not halted [24].
In the case of CSS caused by pathogenic microorganisms, not only must the diagnostic criteria of HLH-2004 be met, but also there must be evidence of an etiological diagnosis (positive tests for the presence of virus or microorganism isolation) [24]. “
References:
23. Alcami, A.; Koszinowski, U.H. Viral Mechanisms of Immune Evasion. Trends Microbiol 2000, 8, 410–418.
24. Yongzhi, X. COVID-19-Associated Cytokine Storm Syndrome and Diagnostic Principles: An Old and New Issue. Emerg Microbes Infect 10, 266–276.
The clinical evidence of covid and HLH should be reflected or summarized in a table that helps the reader understand this possible association.
Following your advice, to illustrate the clinical evidence for covid and HLH, we've added the table below:
Moderately severe to severe
COVID-19
sHLH
prevalence in pregnant women – rarely diagnosed
etiology
viral infection
viral infection in most cases hyperferritinaemia
marker of poor prognosis in COVID-19 patients the most characteristic laboratory result of HLH
treatment
treatment aimed at suppressing the cytokine storm
high mortality
increased levels of IL-6
Table 1. Moderately severe to severe COVID-19 as a form of sHLH. sHLH – secondary haemophagocytic lymphohistiocytosis; COVID-19 – coronavirus disease 2019; HLH - haemophagocytic lymphohistiocytosis; IL-6 – interleukin 6.
One of the problems in the manuscript is if the cytokine storm is present in the context of pregnancy, has this phenomenon been described by other viral infections and thus promoving a pathological effect such as HLH ???
We agree wholeheartedly with the Reviewer`s opinion regarding that it is important to clarify if the cytokine storm is present in the context of pregnancy, has this phenomenon been described by other viral infections and thus promoting a pathological effect such as HLH. Therefore, we have added the following lines:
“The complicated phenomenon of maternal immunological tolerance to foetal antigens enables the development of an antigenically foreign foetus in the uterus despite the existence of the maternal immune system capable of rejection. The maternal immune T-cell profile shifts from the Th1 dominance to the Th2 dominance. Pregnant women are more susceptible to infections and viral infections may be very dangerous for them. It is important to clarify whether the cytokine storm present in the context of pregnancy promotes the development of HLH.”
Finally, future ideas or questions in the field must be mentioned to show whether there is a real relationship of HLH in pregnancy during a COVID-19 infection.
We agree with the Reviewer's opinion that it is important to mention future ideas or questions as to whether there is indeed a relationship between HLH in pregnancy and COVID-19 infection and for this reason, we added the following lines:
“It is important to clarify whether the cytokine storm present in the context of pregnancy promotes the development of HLH. Moreover, future research should focus on elucidating the complex relationship between HLH in pregnancy and
COVID-19 infection. The current state of knowledge and the dilemmas regarding this phenomenon have been described by us above.”
We would like to take this opportunity to thank you for all the valuable and highly perceptive remarks which have definitely made a substantial contribution to the quality of our paper.
Yours faithfully,
Arkadiusz Standyło
Chair and Department of Obstetrics and Perinatology, Medical University of Lublin, 20-090 Lublin, Poland
Tel: +48-81-7244-769
E-mail: [email protected]

Round 2

Reviewer 2 Report

Thank you for making those revisions.  I look forward to reading the article in production.